# Halogen Bonds in 2,5-Dihalopyridine-Copper(I) Halide Coordination Polymers

**DOI:** 10.3390/ma12203305

**Published:** 2019-10-11

**Authors:** Carolina von Essen, Kari Rissanen, Rakesh Puttreddy

**Affiliations:** Department of Chemistry, University of Jyvaskyla, P.O. Box 35, FI-40014 Jyväskylä, Finland; carolina@vonessen.eu (C.v.E.); kari.t.rissanen@jyu.fi (K.R.)

**Keywords:** halogen bond, copper, pyridine, halopyridines, dihalopyridine

## Abstract

Two series of 2,5-dihalopyridine-Cu(I)A (A = I, Br) complexes based on 2-X-5-iodopyridine and 2-X-5-bromopyridine (X = F, Cl, Br and I) are characterized by using single-crystal X-ray diffraction analysis to examine the nature of C2−X2···A–Cu and C5−X5···A–Cu halogen bonds. The reaction of the 2,5-dihalopyridines and Cu(I) salts allows the synthesis of eight 1-D coordination polymers and a discrete structure. The resulting Cu(I)-complexes are linked by C−X···A–Cu halogen bonds forming 3-D supramolecular networks. The C−X···A–Cu halogen bonds formed between halopyridine ligands and copper(I)-bound halide ions are stronger than C−X···X’–C interactions between two 2,5-dihalopyridine ligands. The C5−I5···I–Cu and C5−Br5···Br–Cu halogens bonds are shorter for C2-fluorine than C2-chlorine due to the greater electron-withdrawing power of fluorine. In 2,5-diiodopyridine-Cu(I)Br complex, the shorter C2−I2···Br–Cu [3.473(5) Å] distances are due to the combined polarization of C2-iodine by C2−I2···Cu interactions and *para*-electronic effects offered by the C5-iodine, whilst the long halogen bond contacts for C5−I5···Br–Cu [3.537(5) Å] are indicative that C2-iodine has a less *para*-electronic influence on the C5-iodine. In 2-fluoro-5-X-pyridine-Cu(I) complexes, the C2-fluorine is halogen bond passive, while the other C2-halogens in 2,5-dihalopyridine-Cu(I), including C2-chlorine, participate in halogen bonding interactions.

## 1. Introduction

Supramolecular chemistry utilizes small molecules and non-covalent interactions to self-assembly molecular aggregates with properties that are different from their individual components [1]. The design and construction of materials by self-assembly are largely predestined based on hydrogen bonding interactions due to the small size, easily polarizable and ubiquitous nature of the hydrogen atom [2]. Hydrogen bonding is observed in organic, as well as coordination, compounds. Over the last few years, orthogonal and conceptually similar to hydrogen bonds, halogen bonding has been studied as an additional important non-covalent interaction [3]. The halogen bonding is highly directional and the contact distances between donor and acceptor molecules can be modulated, due to the polarization hierarchy of different halogen atoms, in co-crystals for applications in material science [4] and solid-state research [5]. 

The examination of halogen bonds in coordination compounds is still an underrepresented topic in halogen bond (XB) research [6,7,8,9,10,11,12,13,14,15]. *mono*-Halopyridines are common organic moieties used for exploring halogen bonds in coordination complexes [16,17,18,19]. The N–M (N = nitrogen, M = metal) coordination bond increases the σ-hole strength on the halogen atom that is covalently bound to the core aromatic ring [20,21]. The polarization also leads to the formation of anisotropically distributed electron density on the halogen atom orthogonal to the σ-hole facilitating XB acceptor properties. Two types of halogen bonds, therefore, are typically expected in coordination complexes, C–X···A–M [X = XB donor, A = XB acceptor] and C–X···X′–C [X = XB donor, X′ = XB acceptor] [22]. Square planar complexes of the type [M(*mono*-halopyridine)_2_A_2_] (M = Cu, Ni, Pt, Pd; A = Cl, Br, I) [23,24,25,26,27,28] and co-crystals of the type [(*mono*-halopyridine)H]^+^[MA_n_]^−^ (for example, MA_n_ = CoBr_4_, CoCl_4_, PtCl_6_) [29,30,31,32] have been studied as earlier contributions for halogen bonds in coordination complexes. In X-ray crystal structures, the short X···A distances and C–X···A linear angles are evidence for halogen bonds [33], the geometrical parameters typically utilized to characterize the hydrogen bonding interactions [34].

Our interest in the halogen bonding of coordination complexes led us to utilize 2,5-dihalopyridine to prepare 2:1 ligand:metal ratio [Cu(2,5-dihalopyridine)_2_A_2_] complexes, and to examine the electronic influence of C2-halogens on C5–X5···A–Cu halogen bonds and C5-halogens on C2–X2···A–Cu halogen bonds (A = Cl, Br) [35,36]. The electronic or substituent effects are monitored by measuring the C2–X2···A–Cu and C5–X5···A–Cu distances, and compared with the respective C2–X2···A–Cu and C3–X3···A–Cu distances in [Cu(2-halopyridine)_2_A_2_] and [Cu(3-halopyridine)_2_A_2_] complexes. This strategy also allowed us to rank the XB strengths for C5–X5···A–Cu (A = Cl, Br) halogen bonds, while a mixed order was observed for C2–X2···A–Cu halogen bonds. The X-ray crystal structures of [Cu(2,5-dihalopyridine)_2_A_2_] complexes are discrete 2:1 ligand:metal complexes. Therefore, the electrostatic attractions between externally directed Cu(II)-bound halide and electrophilic halogen of a 2,5-dihalopyridine for C–X···A–M halogen bonds are the major determinants when compared to the crystal packing forces. Consequently, the [Cu(2,5-dihalopyridine)_2_A_2_] complexes were the ideal candidates to understand the electronic effects in C–X···A–M halogen bonds. Although the C2– and C5–halogen polarizations seem to imply the strong dependence on the electronic structure of the pyridine ring, the C–X···A–M halogen bonds could be sensitive to the dimensionality of copper complexes. In this study, the Cu(I) salts provided a means for understanding the nature of C2–X2···A–Cu and C5–X5···A–Cu (A = Br, I) halogen bonds in 2,5-dihalopyridine-Cu(I) coordination polymeric structures.

## 2. Results

Nine Cu(I)-complexes were obtained by mixing a 1:1 molar ratio of six different 2,5-dihalopyridines (1–6, Figure 1) and two copper(I) halides (CuI and CuBr) in a 1:1 ratio of acetonitrile:ethanol mixture whilst heating gently. Slow evaporation of the resulting solutions provided single crystals suitable for X-ray diffraction analysis. Our attempts to grow single crystals of 5-bromo-2-iodopyridine-Cu(I) and 5-bromo-2-chloropyridine-Cu(I) complexes were unsuccessful. The Cu(I)-complexes of ligands **1**–**6** are labelled using the letter “**a**” for 2,5-dihalopyridine-CuI and “**b**” for 2,5-dihalopyridine-CuBr complexes. Five complexes (**1a**, **1b**, **2a**, **3a**, **5a**) form isostructural 1-D coordination polymers, whilst the other three complexes (**2b**, **4b**, and **6b**) show different structures and are described individually.

In complexes **1a**, **1b**, **2a**, **3a**, and **5a**, two Cu(I) ions and two halide atoms form Cu_2_A_2_ (A = I, Br) rhomboid-like units, where the two edges of each rhomboid are shared by two other adjacent rhomboids to yield 1-D coordination polymer ladders (Figure 2a,c,e,g). The Cu(I) ions are coordinated by one pyridine nitrogen and three halide atoms in a tetrahedral geometry with NA_3_ coordination spheres. The pyridine ligands in the ladder are parallel to each other, not forming any π–π interaction between the aromatic rings, indicated by adjacent aromatic centroid-to-centroid distances of 4.30 Å [**1a**], 4.0 Å [**1b**], 4.30 Å [**2a**], 4.20 Å [**3a**], and 4.20 Å [**5a**]. The small aromatic centroid-to-centroid distances further suggested that the interdigitation between 1-D polymers is not feasible. In the crystal lattice, C5-halogens and Cu(I)-bound halides of adjacent 1-D coordination polymer ladders manifest C5–X5∙∙∙A–Cu halogen bonds, as shown in Figure 2b,d,f,h. The higher the electronegativity of the C2-halogen, the shorter the C5–X5∙∙∙I–Cu halogen bonds. For example, the C5–I5∙∙∙I–Cu halogen bond (R_XB_ = 0.92) in **1a** is shorter in comparison to the C5–I5∙∙∙I–Cu halogen bonds (R_XB_ = 0.93) in **2a** due to the higher electron-withdrawing nature of the C2-fluorine when compared to C2-chlorine. The electron-withdrawing effect of C2-halogen is also observed in **5a** and **6a** complexes for C5–Br5∙∙∙I–Cu halogen bonds. The C2-halogens in **1a**, **1b**, **2a**, and **5a** are XB passive, meaning they did not function as XB-donor and acceptors. 

The complex formed by 2-bromo-5-iodopyridine (**3**) and CuI is the exotic structure of the 2-X-5-iodopyridine series (X = F, Cl, Br, and I). Typically, C2–X2∙∙∙I–Cu or C5–X5∙∙∙I–Cu XBs are observed but in this case, C2- and C5-halogens form C2–Br2∙∙∙I5–C5 halogen bonds. The C2-bromine is the XB donor and C5-iodine is the XB acceptor with ∠C2–Br2∙∙∙I5 = 172(1)° (Figure 3). The C2–Br2∙∙∙I5–C5 halogen bond contacts are just below the sum of the van der Waals radii of bromine and iodine atoms (3.83 Å). This could be either due to the less electron-withdrawing power of C2-bromine or the weak donating ability of C5-iodine caused by crystal-packing interactions. The copper(I)-bound iodine ion is XB passive, and only forms C4–H4∙∙∙I–Cu interactions with a distance of 3.10 Å (∠C4–H4∙∙∙I = 148°). Furthermore, the C5-iodines from the adjacent 1-D polymeric structures display weak C5–I5∙∙∙I5′–C5′ interactions at distances of 3.936(4) Å [R_XB_ = 0.99; ∠C5–I5∙∙∙I5′ = ∠C5′–I5′∙∙∙I5 = 146.6(2)].

Single crystals of complex **1b**･ACN were isolated from the bulk sample **1b**. The asymmetric unit contains two crystallographically independent Cu(I) ions, a pyridine ligand and coordinating acetonitrile solvent molecule. Both Cu(I) ions have tetrahedral geometry with a NBr_3_ coordination sphere. However, one Cu(I)-centre is coordinated by pyridine nitrogen and three μ_2_-Br ions while the second Cu(I)-centre is coordinated by acetonitrile nitrogen and three μ_2_-Br ions, as shown in Figure 4a. The CuBr cluster in **1b**･ACN extends into a 1-D polymeric structure, similar to complexes **1a**, **1b**, **2a**, and **5a**, with Cu(I) coordinated pyridine ligands and acetonitrile solvents decorated on the opposite side, as depicted in Figure 4a. The C5-iodine and Cu(I)-bound bromide from adjacent 1-D chains form C5–I5∙∙∙Br–Cu halogen bonds which are shorter in comparison to C5–I5∙∙∙Br–Cu XBs in **1b**. In the crystal packing, the C2-fluorine and acetonitrile *sp* and *sp*^3^ carbon atoms are electrostatically driven closer for C2–F2∙∙∙C(*sp*) [d = 2.790(2) Å, ∠C2–F2∙∙∙C = 145.3(9)°] and C2–F2∙∙∙C(*sp*^3^) [d = 3.076 (2) Å, ∠C2–F2∙∙∙C = 171.2(8)°] short contacts, as shown in Figure 4b.

Complex **4b** crystallizes in the triclinic space group *P*-1 and is a 1-D polymeric chain with a CuBr cluster core different to **1b**. The Cu(I) ions are coordinated in a trigonal planar fashion by two iodine ions and a pyridine nitrogen (Figure 5). Although the C2- and C5-positions in ligand **4** are iodine, which is typically known for its strong XB-donor ability, the C2–I2∙∙∙Br–Cu halogen bonds are shorter than the C5–I5∙∙∙Br–Cu XBs due to additional polarization caused by C2–I2∙∙∙Cu interactions [3.333(5) Å] and the electron-withdrawing C5-halogen. The short C2–I2∙∙∙Cu contacts are similar to C2–Br2∙∙∙Cu interactions in our previously reported [Cu(2,5-dihalopyridine)_2_Br_2_] complexes [31]. 

Complex **6a** crystallizes in the monoclinic space group *P*2_1_/n. The asymmetric unit contains one pyridine ligand, two Cu(I) ions, and two iodide ions. The CuI and nitrogen in pyridine ligands of **6a** form a 1-D honeycomb-like coordination polymer with Cu_6_I_6_N_2_ nodes, as shown in Figure 6. Both Cu(I) ions are tetrahedrally coordinated. One Cu(I) is coordinated by one pyridine nitrogen, one μ_3_- and two μ_4_-iodide ions, and the other Cu(I) is coordinated by two μ_3_- and two μ_4_-iodide ions, leading to Cu(I)-centers with NI_3_ and I_4_ coordination spheres, respectively. The pyridine ligands are decorated parallel to each other above and below the 1-D chain, with centroid-to-centroid distances of 4.20 Å. The C5-bromine σ-hole and nucleophilic μ_3_-iodide ion form C5–I5∙∙∙I–Cu halogen bonds whilst the C5-bromine anisotropic electron ′′belt′′ at the orthogonal directions and C2-bromide σ-hole form a C5–I5∙∙∙I2–C2 type halogen bond. The μ_4_-iodide ion is XB passive.

Complex **2b** is the only example of a discrete Cu(I)-complex. The asymmetric unit consists of three crystallographically independent pyridine ligands (Py1-Py3) and two Cu(I) ions with a tetrahedral and a trigonal planar coordination geometry, respectively. In the molecular packing, the discrete structures exhibit several halogen bonds and halogen∙∙∙halogen interactions; the respective distances and bond parameters are given in Table 1. Within the asymmetric unit, the C2-chlorine (Py1) and C5-iodine (Py3) attract electrostatically to form weak (Py1)C2–Cl2∙∙∙I5–C5(Py3) halogen bonds with distances of 3.688(3) [∠C2–Cl2∙∙∙I5 = 176.8(4); R_XB_ = 0.99]. The C2-chlorine (Py1) is an XB-donor to the C5-iodine (Py3) XB-acceptor (Figure 7b). These halogen-bond contacts are weak, just below the sum of the van der Waals radii of Cl- and I-atoms [3.73 Å]. The C5-iodine substituents of Py1 and Py2 form relatively strong halogen bonds between two crystallographically independent copper-bound bromines (Figure 7b,c). The C5-iodine of Py3 exhibits two other halogen∙∙∙halogen contacts to C2-chlorines of (Py3′’) [R_XB_ = 0.95] and (Py2′) [R_XB_ = 0.98] to neighboring complexes, as depicted in Figure 7c. The strongest interactions are observed between C5-iodine substituents and nucleophilic Cu(I)-bound bromine ions.

## 3. Conclusions

The present study shows the application of rarely used 2,5-dihalopyrdine ligands to prepare copper(I)-coordination polymers that are stabilized by halogen bond interactions in the solid-state. The copper(I) complexes extend into 3-D supramolecular networks through C2–X2···A–Cu and C5–X5···A–Cu halogen bonds (X = Cl, Br, I; and A = Br, I) between the X2- or X5-halogen substituent (donor) and the copper(I)-bound halide anion (acceptor). With the exception of the C2-fluorine, all C2- and C5-halogen substituents, including the more electronegative C2-chlorine, participate in halogen bond interactions. Strong halogen bonds are formed between the C5-halogen substituent and the nucleophilic halide ions coordinated to copper(I), with C5-halogen always acting as XB donor and the halide anion as an XB acceptor. The flexible copper(I) coordination sphere allows the 2,5-dihalopyridine halogen substituents to function as a halogen bond acceptor for C–X···X’–C interactions, in addition to C2–X2···A–Cu, C5–X5···A–Cu halogen bonds. This feature is not promoted by copper(II) in our previously reported discrete structures of [Cu(2,5-dihalopyridine)_2_A_2_]. In 2-X-5-iodopyridine-Cu(I)A complexes (X = F, Cl, and A = Br, I), the C5–X5···A–Cu halogen bonds are shorter for C2-fluorine than C2-chlorine due to the higher electron-withdrawing effect of fluorine, similar to our previously reported [Cu(2,5-dihalopyridine)_2_A_2_] complexes. Of the four ligands, 2-chloro-5-iodopyridine, 2-bromo-5-iodopyridine, 2,5-diiodopyridine, and 2,5-dibromopyridine, only C2-iodine of 2,5-diiodopyridine displays C2–I2···Cu short contacts. This effect depends on the coordination geometry of the Cu(I)-center and is widely observed in square planar [Cu(2,5-dihalopyridine)_2_A_2_] complexes. 

## 4. Experimental Section

**General information**: In the crystallization experiments, all used solvents were of reagent grade and were used as received from the supplier. The pyridine ligands, 2-fluoro-5-iodopyridine (**1**), 2-chloro-5-iodopyridine (**2**), 2-bromo-5-iodopyridine (**3**), 2,5-diiodopyridine (**4**), 5-bromo-2-fluoropyridine (**5**), and 2,5-dibromopyridine (**6**) are commercially available (TCI Chemicals Europe), the CuBr and CuI salts were purchased from Sigma Aldrich. 

**General synthesis of complexes 1a–6a**: The solid 2,5-dihalopyridine (0.1046 mmol) was added to the solution of CuA (A = Br, I) (0.1046 mmol) in acetonitrile/ethanol (2.0 mL). If needed, the solutions were heated to dissolve the components. Single-crystals suitable for X-ray diffraction analysis were obtained by slow evaporation of the corresponding solutions.

**Crystal structure determination**: The X-ray data for **1b**, **2a**, **2b**, **4b** and **5a** were obtained by using a Bruker-Nonius Kappa CCD diffractometer with an APEX-II CCD detector utilizing graphite-monochromated Mo-Kα (λ = 0.71073 Å) radiation. Those of **1b**･ACN, **3a**, and **6a** data were collected with a Rigaku Oxford Diffraction SuperNova instrument with an EoS CCD detector. The used Mo-Kα (λ = 0.71073 Å) radiation was monochromatized using multi-layer optics. The data for **1****a** was collected using the Rigaku SuperNova dual-source Oxford diffractometer equipped with an Atlas detector using mirror-monochromated Cu-Kα (λ = 1.54184 Å) radiation. The *CrysAlisPro* program and Gaussian face-index absorption correction method were used for data collection and reduction for **1a**, **1b**･ACN, **3a**, and **6a**. For the data collection and reduction for **1b**, **2a**, **2b**, **4b** and **5a**, the programs COLLECT and HKL DENZO AND SCALEPACK [38] were used. The intensities were multi-scan absorption-corrected using SADABS [39]. Direct methods (SHELXS) [40] and full-matrix least squares on F^2^ using the *OLEX2* software [41] with SHELXL-2013 module [40] were used for all structures.

Crystal data for **1a**: CCDC-1951451, C_5_H_3_CuFI_2_N, M = 413.42, colorless needle, 0.373 × 0.032 × 0.026 mm^3^, triclinic, space group *P*-1, a = 4.2414(9) Å, b = 8.9099(15) Å, c = 11.1529(17) Å, α = 77.445(14)°, β = 82.242(15)°, γ = 85.595(15)°, V = 407.14(13) Å^3^, Z = 2, Dc = 3.372 g/cm^3^, F000 = 368, μ= 62.874 mm^−1^, T = 120.01(10) K, θ_max_ = 66.728°, 3438 total reflections, 1299 with Io > 2σ(Io), R_int_ = 0.0603, 1430 data, 79 parameters, 0 restraints, GooF = 1.055, R = 0.0613 and wR= 0.0644 [Io > 2σ(Io)], R = 0.1586 and wR= 0.1639 (all reflections), 3.472 < d∆ρ < −2.035 e/Å^3^.

Crystal data for **1b**: CCDC-1951453, C_5_H_3_BrCuFIN, M = 366.43, colorless plate, 0.14 × 0.014 × 0.06 mm^3^, monoclinic, space group *P*2_1_/n, a = 8.7766(18) Å, b = 4.0769(8) Å, c = 21.701(4) Å, α = 90°, β = 100.81(3)°, γ = 90°, V = 762.7(3) Å^3^, Z = 4, Dc = 3.191 g/cm^3^, F000 = 664, μ= 12.083 mm^−1^, T = 170.0(1) K, θ_max_ = 24.994°, 5261 total reflections, 1154 with Io > 2σ(Io), R_int_ = 0.0603, 1329 data, 91 parameters, 12 restraints, GooF = 1.272, R = 0.0913 and wR = 0.2320 [Io > 2σ(Io)], R = 0.1006 and wR = 0.2352 (all reflections), 2.593 < d∆ρ < −2.201 e/Å^3^.

Crystal data for **1b**･ACN: CCDC-1951452, C_7_H_6_Br_2_Cu_2_FIN_2_, M = 550.94, colorless needle, 0.07 × 0.03 × 0.021 mm^3^, monoclinic, space group *P*2_1_/n, a = 3.9980(2) Å, b = 8.7053(5) Å, c = 35.319(3) Å, α = 90°, β = 91.118(6)°, γ = 90°, V = 1229.00(13) Å^3^, Z = 4, Dc = 2.978 g/cm^3^, F000 = 1008, μ= 12.454 mm^−1^, T = 120.0(1) K, θ_max_ = 29.879°, 8335 total reflections, 2393 with Io > 2σ(Io), R_int_ = 0.0845, 3229 data, 138 parameters, 6 restraints, GooF = 1.202, R = 0.0999 and wR = 0.1436 [Io > 2σ(Io)], R = 0.1363 and wR = 0.1551 (all reflections), 1.996 < d∆ρ < −3.084 e/Å^3^.

Crystal data for **2a**: CCDC-1951454, C_5_H_3_ClCuI_2_N, M = 429.87, colorless plate, 0.19 × 0.11 × 0.08 mm^3^, monoclinic, space group *P*2_1_/c, a = 4.2871(9) Å, b = 18.208(4) Å, c = 11.464(2) Å, α = 90°, β = 99.46(3)°, γ = 90°, V = 882.7(3) Å^3^, Z = 4, Dc = 3.235 g/cm^3^, F000 = 768, μ= 9.696 mm^−1^, T = 170.0(1) K, θ_max_ = 25.248°,5962 total reflections, 1343 with Io > 2σ(Io), R_int_ = 0.0406, 1592 data, 91 parameters, 0 restraints, GooF = 1.024, R = 0.0259 and wR = 0.0453 [Io > 2σ(Io)], R = 0.1006 and wR = 0.0478 (all reflections), 0.737 < d∆ρ < −0.610 e/Å^3^.

Crystal data for **2b**: CCDC-1951455, C_15_H_9_Br_2_Cl_3_Cu_2_I_3_N_3_, M = 1005.20, colorless needle, 0.06 × 0.028 × 0.027 mm^3^, triclinic, space group *P*-1, a = 8.4309(17) Å, b = 9.4460(19) Å, c = 16.953(3) Å, α = 100.81(3)°, β = 97.74(3)°, γ = 112.52(3)°, V = 1192.9(5) Å^3^, Z = 2, Dc = 2.799 g/cm^3^, F000 = 916, μ= 9.359 mm^−1^, T = 170.0(1) K, θ_max_ = 25.250°, 7687 total reflections, 3498 with Io > 2σ(Io), R_int_ = 0.0531, 4270 data, 253 parameters, 0 restraints, GooF = 1.018, R = 0.0541 and wR = 0.1348 [Io > 2σ(Io)], R = 0.0674 and wR = 0.1432 (all reflections), 2.097 < d∆ρ < −1.399 e/Å^3^.

Crystal data for **3a**: CCDC- 1951456, C_5_H_3_BrCuI_2_N, M = 474.33, colorless needle, 0.181 × 0.067 × 0.048 mm^3^, monoclinic, space group *P*2_1_/c, a = 4.1815(3) Å, b = 14.7272(14) Å, c = 14.8718(12) Å, α = 90°, β = 91.173(7)°, γ = 90°, V = 915.64(13) Å^3^, Z = 4, Dc = 3.441 g/cm^3^, F000 = 840, μ= 13.420 mm^−1^, T = 170.0(1) K, θ_max_ = 25.248°, 4840 total reflections, 1505 with Io > 2σ(Io), R_int_ = 0.0432, 1646 data, 49 parameters, 0 restraints, GooF = 1.248, R = 0.1279 and wR = 0.3369 [Io > 2σ(Io)], R = 0.1319 and wR = 0.3384 (all reflections), 5.704 < d∆ρ < −2.773 e/Å^3^.

Crystal data for **4b**: CCDC-1951457, C_5_H_3_BrCuI_2_N, M = 474.33, colorless needle, 0.12 × 0.11 × 0.09 mm^3^, triclinic, space group *P*-1, a = 4.0630(8) Å, b = 8.9680(18) Å, c = 12.931(3) Å, α = 74.44(3)°, β = 85.36(3)°, γ = 84.90(3)°, V = 451.31(17) Å^3^, Z = 2, Dc = 3.491 g/cm^3^, F000 = 420, μ= 13.613 mm^−1^, T = 170.0(1) K, θ_max_ = 24.998°, 2593 total reflections, 1381 with Io > 2σ(Io), R_int_ = 0.0400, 1513 data, 80 parameters, 36 restraints, GooF = 1.220, R = 0.0930 and wR = 0.2613 [Io > 2σ(Io)], R = 0.1002 and wR = 0.2652 (all reflections), 4.442 < d∆ρ < −2.380 e/Å^3^.

Crystal data for **5a**: CCDC-1951458, C_5_H_3_BrCuFIN, M = 366.43, colorless plate, 0.409 × 0.071 × 0.043 mm^3^, monoclinic, space group *P*2_1_/n, a = 8.9290(18) Å, b = 4.1740(8) Å, c = 21.620(4) Å, α = 90°, β = 101.57(3)°, γ = 90°, V = 789.4(3) Å^3^, Z = 4, Dc = 3.083 g/cm^3^, F000 = 664, μ= 11.675 mm^−1^, T = 170.0(1) K, θ_max_ = 28.363°, 6126 total reflections, 1630 with Io > 2σ(Io), R_int_ = 0.0343, 1932 data, 91 parameters, 0 restraints, GooF = 1.164, R = 0.0354 and wR = 0.0795 [Io > 2σ(Io)], R = 0.0464 and wR = 0.0834 (all reflections), 1.100 < d∆ρ < −0.862 e/Å^3^.

Crystal data for **6a**: CCDC-1951459, C_5_H_3_Br_2_Cu_2_I_2_N, M = 617.78, colorless needle, 0.14 × 0.09 × 0.08 mm^3^, monoclinic, space group *P*2_1_/n, a = 4.19570(10) Å, b = 13.8695(7) Å, c = 18.9218(7) Å, α = 90°, β = 90.509(3)°, γ = 90°, V = 1101.06(7) Å^3^, Z = 4, Dc = 3.727 g/cm^3^, F000 = 1096, μ= 16.674 mm^−1^, T = 170.03(10) K, θ_max_ = 25.240°, 6988 total reflections, 1655 with Io > 2σ(Io), R_int_ = 0.0457, 1982 data, 109 parameters, 0 restraints, GooF = 1.019, R = 0.0294 and wR = 0.0615 [Io > 2σ(Io)], R = 0.0390 and wR = 0.0676 (all reflections), 0.927 <d∆ρ < −0.888 e/Å^3^.

## Figures and Tables

**Figure 1 materials-12-03305-f001:**
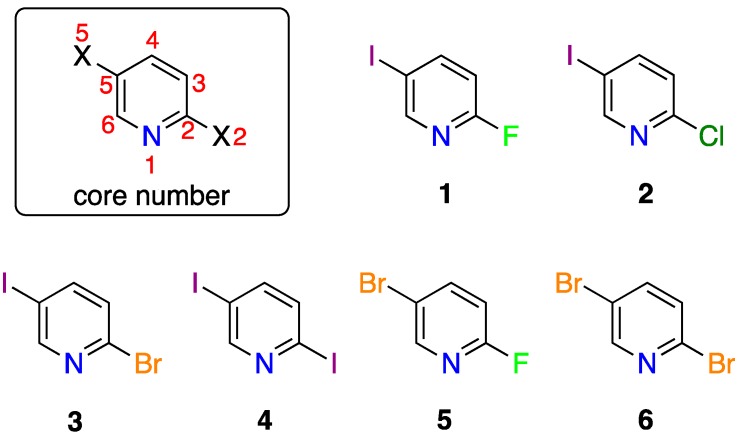
Chemical structures of 2,5-dihalopyridines: 2-fluoro-5-iodopyridine (**1**), 2-chloro-5-iodopyridine (**2**), 2-bromo-5-iodopyridine (**3**), 2,5-diiodopyridine (**4**), 2-fluoro-5-bromopyridine (**5**), and 2,5-dibromopyridine (**6**).

**Figure 2 materials-12-03305-f002:**
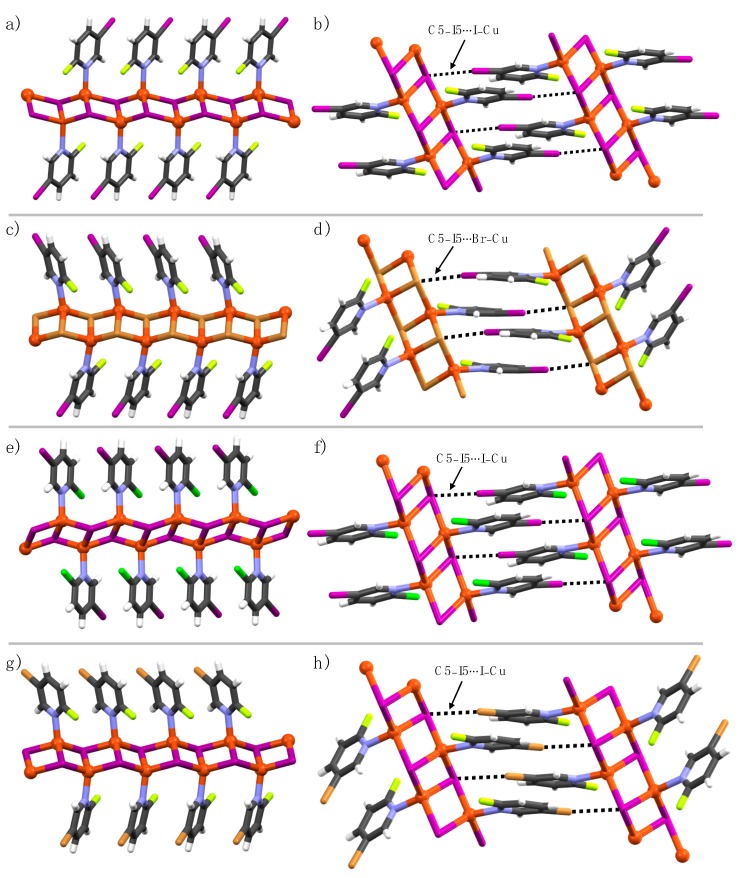
1-D Polymeric view of (**a**) **1a**, (**b**) **1b**, (**c**) **2a**, and (**d**) **5a**, and (**e**–**h**) their respective section of crystal packing structures displaying halogen bonding interactions (black dotted lines).

**Figure 3 materials-12-03305-f003:**
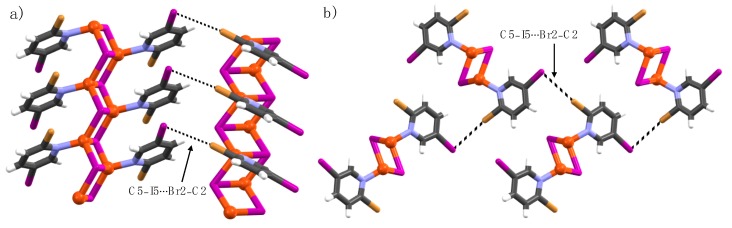
(**a**,**b**) X-Ray crystal structure of **3a** viewed from two different directions. Halogen bonds are depicted by using black dotted lines.

**Figure 4 materials-12-03305-f004:**
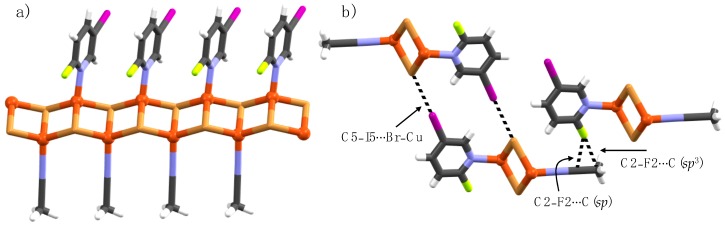
(**a**) 1-D Polymer view of **1a**･ACN, and (**b**) section of crystal packing viewing halogen bonds and C2–F2∙∙∙C interactions (black dotted lines).

**Figure 5 materials-12-03305-f005:**
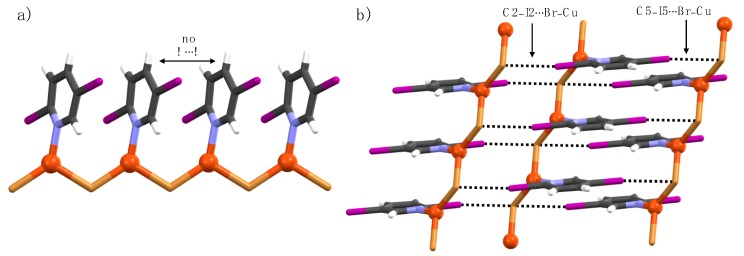
(**a**) X-Ray crystal Structure of **4b**, and (**b**) section of 1-D polymer chains in crystal packing displaying halogen bonds (black dotted lines).

**Figure 6 materials-12-03305-f006:**
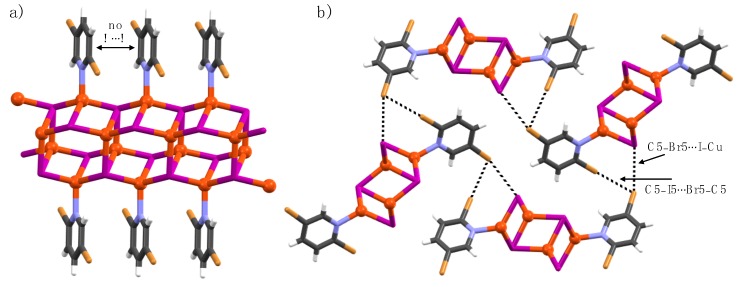
(**a**) X-Ray crystal structure of **6a**, and (**b**) section of crystal packing displaying halogen bonds. Halogen bonds are depicted by black dotted lines.

**Figure 7 materials-12-03305-f007:**
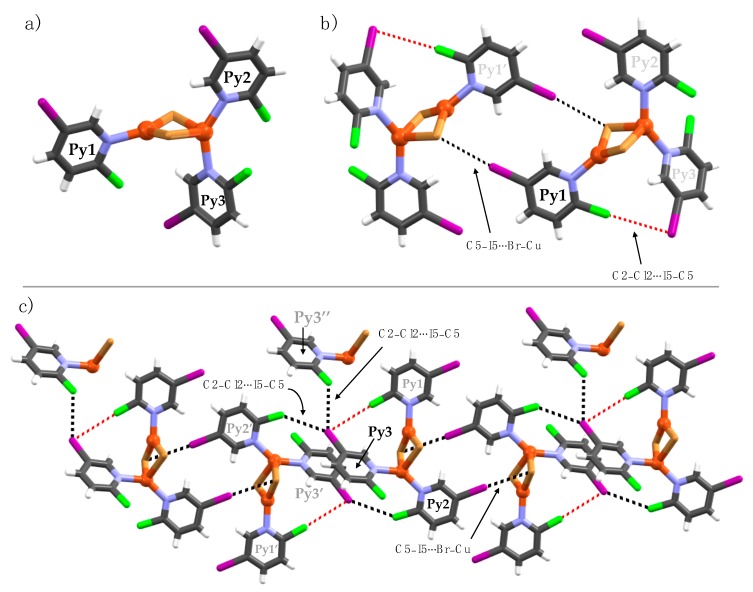
Crystal structure of **2b**. Halogen bonds are depicted as black and red dotted lines focusing on (**a**) Py1, (**b**) Py2 and (**c**) Py3.

**Table 1 materials-12-03305-t001:** The solid-state X-ray crystal structure C–X∙∙∙A–Cu bond parameters of 2,5-dihalopyridine-Cu(I) complexes.

Copper(I) Iodide Complexes				
Complex	Motif	d(C5–X5∙∙∙Y) [Å]	∠ (C5–X5∙∙∙Y) [°]	R_XB_^a^
**1a**	C5–I5∙∙∙I–Cu	3.656(2)	174.5(4)	0.92
**2a**	C5–I5∙∙∙I–Cu	3.679(1)	176.1(1)	0.93
**3a**	C5–I5∙∙∙Br2–C2	3.796(5)	172(1)	0.99
**5a**	C5–Br5∙∙∙I–Cu	3.627(2)	173.2(2)	0.95
**6a**	C5–Br5∙∙∙I–Cu	3.678 (3)	160.6(5)	0.96
	C2–Br2∙∙∙Br5–C5	3.609(2)	177.2(4)	0.98
**Copper(I) Bromide Complexes**				
**1b**	C5–I5∙∙∙Br–Cu	3.503(3)	174.0(7)	0.91
**1b·ACN**	C5–I5∙∙∙Br–Cu	3.470(2)	176.3(4)	0.91
**2b**	(Py1)C5–I5∙∙∙Br–Cu	3.434(2)	175.4(3)	0.90
	(Py2)C5–I5∙∙∙Br–Cu	3.515(2)	174.3(3)	0.92
	(Py1)C2–Cl2∙∙∙I5–C5(Py3)	3.688(3)	176.8(4)	0.99
	(Py3)C5–I5∙∙∙Cl2–C2(Py3″)	3.556(3)	135.6(4)	0.95
	(Py3)C5–I5∙∙∙Cl2–C2(Py2′)	3.659(4)	142.9(4)	0.98
**4b**	C2–I2∙∙∙Br–Cu	3.473(5)	172(1)	0.91
	C5–I5∙∙∙Br–Cu	3.537(5)	169(1)	0.92

^a^ The normalized interaction ratio, R_XB_, is defined as (R_XB_ = d_XB_/X_vdw_+B_vdw_), where d_XB_ [Å] is the distance between the donor atom (X) and the acceptor atoms (B) and divided by the sum of vdW radii [Å] of X and B. The van der Waals radii determined by Bondi were used to calculate the R_XB_ values [37].

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
