# Peer review of "Halogen Bonds in 2,5-Dihalopyridine-Copper(I) Halide Coordination Polymers"

_materials, 2019, doi:10.3390/ma12203305_

Round 1

Reviewer 1 Report

This is a well-executed body of structural correlations in a series of crystalline materials whose design successfully rely on halogen-bonding interactions. This is a valuable addition that will attract attention from the practitioners of the field. 

Author Response

The reviewer’s comments and revision suggestions are shown in italics font and our response in the bold-face font.

Reviewer 1

This is a well-executed body of structural correlations in a series of crystalline materials whose design successfully rely on halogen-bonding interactions. This is a valuable addition that will attract attention from the practitioners of the field.

Response:

We thank the referee for his/her positive comments of our work and supporting the manuscript for publication in Materials.

Reviewer 2 Report

The paper "Halogen bonds in 2,5-Dihalopyridine-Copper(I) Halide Coordination Polymers" by Puttreddy and coworkers is technically sound and interesting for the field of halogen bond.

The characterization of the new structures here obtained is satisfactory, even if not multi-technique, as it could have been to gain even more interest. The differences among the various structures are remarkable, even if I do not agree on the final sentence 

" C2−X2∙∙∙A–Cu and C5−X5∙∙∙A–Cu halogen bonds are an important crystal engineering tool"

it would be true if the final structure could be predicted, somehow, but here I do not see any correlation between the crystal structures and the properties of the starting materials. If there is some correlation, and most importantly if the structures show a rationale that make them expected (even a posteriori), the authors should better underline it.

Author Response

The reviewer’s comments and revision suggestions are shown in italics font and our response in bold-face font.

Reviewer 2

The paper "Halogen bonds in 2,5-Dihalopyridine-Copper(I) Halide Coordination Polymers" by Puttreddy and coworkers is technically sound and interesting for the field of halogen bond.

The characterization of the new structures here obtained is satisfactory, even if not multi-technique, as it could have been to gain even more interest. The differences among the various structures are remarkable, even if I do not agree on the final sentence 

" C2−X2∙∙∙A–Cu and C5−X5∙∙∙A–Cu halogen bonds are an important crystal engineering tool"

it would be true if the final structure could be predicted, somehow, but here I do not see any correlation between the crystal structures and the properties of the starting materials. If there is some correlation, and most importantly if the structures show a rationale that make them expected (even a posteriori), the authors should better underline it.

Response:

We thank the referee for his/her positive comments on our manuscript. We have made the suggested changes, by removing the sentence " C2−X2∙∙∙A–Cu and C5−X5∙∙∙A–Cu halogen bonds are an important crystal engineering tool" in the revised manuscript.

Reviewer 3 Report

In this work, Rissanen et al. expand the series of complexes with halogen-substituted pyridines (X-Py) which revealed to be very promising building blocks for halogen bonding-assisted formation of supramolecular frameworks. All results are novel, and scientific quality of material is high, so there are no doubts that this article deserves being published in Materials. I have just two recommendations:

1) There are three recent works where the role of X-Py-type cations on XB formation was discussed, so maybe these articles are worth mentioning here. Those are:

a) 10.1016/j.poly.2019.03.041

b) 10.1039/c8ce01749b

c) 10.1002/chem.201703747

2) Although the overall impression of this MS is very good, I think that it could become even better if authors additionally report estimations of XB energies (via DFT calculations, for example, as described here: 10.1039/c7ce01487b). However, this is just recommendation; such addition is, of course, not mandatory.

Author Response

The reviewer’s comments and revision suggestions are shown in italics font and our response in the bold-face font.

Reviewer 3

In this work, Rissanen et al. expand the series of complexes with halogen-substituted pyridines (X-Py) which revealed to be very promising building blocks for halogen bonding-assisted formation of supramolecular frameworks. All results are novel, and scientific quality of material is high, so there are no doubts that this article deserves being published in Materials. I have just two recommendations:

Response:

We thank the referee for his/her positive review on our manuscript. We are happy the reviewer thinks the manuscript is suitable for publication in Materials.

1) There are three recent works where the role of X-Py-type cations on XB formation was discussed, so maybe these articles are worth mentioning here. Those are:

a)1016/j.poly.2019.03.041 b)1039/c8ce01749b c)1002/chem.201703747

Response: We have made the suggested changes, by including the three references in the revised manuscript.

2) Although the overall impression of this MS is very good, I think that it could become even better if authors additionally report estimations of XB energies (via DFT calculations, for example, as described here: 10.1039/c7ce01487b). However, this is just recommendation; such addition is, of course, not mandatory.

Response: We are currently performing DFT computational studies for a follow-up that is based on ‘’Halogen bonds in 2,5-Dihalopyridine-Copper(I)/Cu(II) complexes’’. This work should provide more detailed information.